# Evaluating F2 region long term trends using the IRI model: A feasible approximation for experimental trends?

Bruno S. Zossi[1,2], Trinidad Duran[3,4], Franco D. Medina[1,2], Blas F. de Haro Barbas[1,2], Yamila Melendi[3,4], Ana G. Elias[1,2]

[1]INFINOA, CONICET-UNT, Tucuman, 4000, Argentina
[2]Laboratorio de Ionosfera, Atmosfera Neutra y Magnetosfera (LIANM), Facultad de Ciencias Exactas y Tecnología (FACET), Universidad Nacional de Tucumán (UNT), Tucuman, 4000, Argentina
[3]Departamento de Física, Universidad Nacional del Sur (UNS), Bahía Blanca, 8000, Argentina
[4]Instituto de Física del Sur (CONICET-UNS), Bahía Blanca, 8000, Argentina

*Correspondence to*: Ana G. Elias (aelias@herrera.unt.edu.ar)

**Abstract.** The International Reference Ionosphere (IRI) is a widely used empirical ionospheric model based on observations from a worldwide network of ionospheric stations. Therefore, it would be reasonable to expect it to capture long-term changes in key ionospheric parameters, such as foF2 and hmF2 linked to trend forcings like greenhouse gases increasing concentration and the Earth's magnetic field secular variation. Despite the numerous reported trends in foF2 and hmF2 derived from experimental data and model results, there are inconsistencies that require continuous refinement of trend estimation methods and regular data updates. This ongoing effort is crucial to address the difficulties posed by the weak signal-to-noise ratio characteristic of ionospheric long-term trends. Furthermore, the experimental verification of these trends remains challenging, primarily due to time and spatial coverage limitations of measured data series. Achieving these needs for long-term trend accurate detection requires extensive global coverage and resolution of ionospheric measurements together with long enough periods spanning multiple solar cycles to properly filter out variations of shorter term than the sought trend. Considering these challenges, IRI-modeled foF2 and hmF2 parameters offer a valuable alternative for assessing trends and obtaining a first approximation of a plausible global picture representative of experimental trends. This work presents these global trend patterns considering the period 1960-2022 using the IRI-Plas 2020 version, which are consistent with other model predictions. While IRI takes explicitly into account the Earth's magnetic field variations, the increasing in the concentration of greenhouse gases appears indirectly through the IG index which is derived from ionospheric measurements. F2-region trends induced by the first mechanism should be important only around the magnetic equator at the longitudinal range with the strongest displacement, and negligible out of this region. Conversely, trends induced by the greenhouse effect, which are the controversial ones, should be dominant away from the geomagnetic equator and should globally average to negative values in both cases: foF2 and hmF2. Effectively, these negative global means are verified by trends based on IRI-Plas, even though not for the correct reasons in the hmF2 case. In addition, a verification was performed for more localized foF2 trends values,

considering data from 9 mid-latitude stations, and a reasonable level of agreement was observed. It is concluded that IRI model
can be a valuable tool for obtaining preliminary approximations of the Earth's magnetic field induced long-term changes in
foF2 and hmF2, and of experimental trends only in the foF2 case. The latter does not hold for hmF2, even if the trends obtained
are close to the expected values.

## 1 Introduction

The International Reference Ionosphere (IRI) (Bilitza et al., 2022) is an empirical model based on observations from diverse
sources. Therefore, it is reasonable to expect it to reflect, to some extent, the long-term trends observed in key ionospheric
parameters such as the F2 region critical frequency, foF2, and the electron density peak height, hmF2. These trends, in
timescales of decades to a century, are theoretically expected as a consequence of trends in certain ionospheric forcings, such
as the increasing greenhouse gases concentration and the Earth's magnetic field secular variation, among others (Lastovicka,
2017, 2021a).
There are countless foF2 and hmF2 reported trends based on experimental data, which combined with model results, led to a
global scenario of trends with the main forcing being the increasing greenhouse gases concentration over the last decades
(Lastovicka, 2017, 2021a). However, several inconsistencies remain to date that require a permanent update of data and
refinement of the trend estimation methods in order to improve the signal-to-noise ratio that is extremely weak in the case of
ionospheric long-term trends. Additionally, experimental verification is still far from being achieved mainly due to two
reasons: the limited time span and sparse spatial coverage of measured data. The time length should cover at least two complete
solar cycles in order to efficiently filter out this variability that is essential for detecting long-term trends. Moreover, the
ionosphere presents other challenges that need extensive series in order to properly identify and analyze trends. Regarding the
spatial coverage, it should be global and with enough resolution so as to detect other forcings interfering with the expected
trends whose intensity depend on location. This is the case, for example, of Earth's magnetic field secular variation effect on
the ionosphere which seems more prominent close to the geomagnetic equator in some longitudinal ranges (Cnossen, 2020;
Elias et al., 2022). Given the difficulty of achieving these two requirements, we found it useful to evaluate trends from IRI
modeled foF2 and hmF2 parameters and to analyze their usefulness as a reliable approximation of experimental trends.
This research initially focuses on presenting the trends spanning the entire planet. These trends are derived for foF2 and hmF2,
which are among the most significant ionospheric parameters (Cander, 2019). They are calculated following the same
methodology applied to experimental data involving the simplest solar activity filtering approach. Furthermore, a comparative
analysis is conducted between the trend values obtained from the IRI model and experimental trends in order to assess their
accuracy. The continued refinement and updating of ionospheric trend estimation methods from data and models, together
with data collection efforts, are essential for improving our understanding of the underlying factors driving long-term changes
in ionospheric parameters and their potential impacts on the diverse systems affected, such as communication and navigation
systems.

This study is structured as follows: Section 2 provides an overview of the IRI model used. Sections 3 and 4 outline the methodology to derive global trends from IRI and to make a comparative analysis between these trends and experimental data of nine selected stations, respectively. The results are presented in Section 5, followed by a comparison with trends derived from a general circulation model in Section 6, and the discussions and conclusions in Sections 7 and 8.

## 2 On some aspects of the IRI model

The IRI model is an observation-based climatological standard model of the ionosphere that is widely used for several purposes, including the prediction of ionospheric behaviour useful for communication and global positioning systems (Gulyaeva and Bilitza, 2012). The model is designed to provide vertical profiles of the main ionospheric parameters for any location over the globe, hours, seasons, and levels of solar activity, representing monthly mean conditions based on experimental evidence. Even though the improvement of the IRI representation of ionospheric parameters, including those selected in this study, still remains a challenge for the IRI Project, and despite its empirical nature and the potential for ongoing improvements, we choose to examine its suitability in estimating F2-region long-term trends.

Since its first edition in 1969 the IRI model has been steadily improved with newer data and with better mathematical descriptions of global and temporal variation patterns. A large number of independent studies have validated the IRI model in comparisons with direct and indirect ionospheric measurements not used in the model development (Gulyaeva and Bilitza, 2012; Bilitza et al., 2022).

In this study, we used an IRI adaptation, IRI-Plas, that has been modified to include the plasmasphere, extending the model up to 20,000 km (Gulyaeva et al., 2011). While traditional IRI versions use a given solar activity proxy, such as IG for foF2, to estimate variations in ionospheric parameters associated to the solar activity quasi-decadal cycle, IRI-Plas allows selecting between 8 different solar proxies, and among them the MgII index (core-to-wing ratio derived from the Mg II doublet at 280 nm). Since we chose this solar activity proxy for the filtering step before trend estimation, we decided to use this IRI version. The IRI-Plas model from Izmiran (Moscow, Russia) was used, available at https://www.izmiran.ru/ionosphere/weather/.

According to IRI general specifications, long-term variations linked to changes in the geomagnetic field are expected since IRI uses the IGRF model to specify magnetic poles and equator, as well as the modified dip latitude, which is an input for foF2 and hmF2 interpolation procedures. Thus, trends due to the magnetic field changes, which are stronger near to geomagnetic poles and equator, may arise from IRI mathematical interpolation coefficients, which ultimately depend on magnetic inclination. Since these changes are extremely small away from the geomagnetic equator, trends observed in other regions could be attributed to additional sources.

A key aspect in the present study is how IRI determines F2 parameters for a given location. To begin, foF2 is obtained from CCIR (Consultative Committee on International Radio) maps that are based on a procedure of numerical mapping of a set of coefficients (CCIR Atlas of Ionospheric Characteristics, 1991) determined from a fitting to observed monthly median foF2 data from a worldwide network of ionosonde stations (~150 in total). From these coefficient maps, IRI reproduces the diurnal,

seasonal and solar activity variation of foF2 in terms of latitude and longitude through Fourier time series. First, there is a set
of functions in terms of geographic coordinates and the modified dip latitude used to describe the variation of the Fourier
coefficients for a given number of harmonics defining the diurnal variation. Then, the seasonal variation is taken into account
through a set of these coefficients (988 in total) for every month of the year. Finally, the solar activity dependence is considered
by having all these monthly coefficients for two different activity levels: $IG_{12}=0$ and $IG_{12}=100$. From a linear fit between these
two extremes, the harmonic coefficients for any solar activity level can be estimated. IG was originally computed using 13
globally distributed ionosonde stations that included two of the 9 stations here analyzed: Kokubunji and Canberra (Liu et al.,
1983). The distribution of these stations was a compromise between good global coverage and reliable long operating
ionosonde stations. Due to station closings and data unavailability, the number of stations used in IG has decreased to four,
but still includes the two stations used in the present study (Brown et al., 2018). Therefore, this proxy, being obtained from
ionospheric measurements, involves foF2 variations not covered by a solar index.
Specifically, when a given solar proxy is selected among the IRI-Plas 8 options, it is automatically converted to other related
indices used by the different modules' procedures (Gulyaeva et al., 2018). In this way, foF2 interannual variation is determined
by $IG_{12}$, since this index finally defines the CCIR coefficient values.
In the case of hmF2, we consider the default option, which corresponds to the AMTB-2013 model (standing for Altadill-
Magdaleno-Torta-Blanch) (Altadill et al., 2013). This model is based on quiet ionosphere data from 26 digisondes collected
between 1998 and 2006. The monthly averages of the global hmF2 variations are represented by spherical harmonics including
modified dip latitude and longitude for two selected levels of $Rz_{12}$ (0 and 100, as in the case of IG). The interannual variation
of hmF2 is defined then by $Rz_{12}$ since, for a given date, hmF2 is obtained from a linear fit of the spherical harmonic coefficients
between $Rz_{12}=0$ and 100 particularized for the corresponding $Rz_{12}$ value. The same procedure is applied in the cases of the
other two options for hmF2 modeling. Thus, the proxy used in this case, unlike foF2 case, is only reflecting solar activity
variability. Nevertheless, we include its long-term trend analysis considering that the correlation between IG and Rz is higher
than 0.99, and that for a given location and hour, foF2 and hmF2 interannual variation highly correlates. Moreover, IG
correlates the highest with Rz exceeding 0.99 along the period 1960-2022. The linear correlation between IG and MgII, F10.7
and Lyman-α, for example, are 0.975, 0.985 and 0.970 respectively.

## 3 Methodology to assess F2-region trends and spatial variation patterns based on IRI


To assess foF2 and hmF2 trends, monthly values were obtained first from IRI-Plas. This model was run over a 5°×10° latitude-
longitude grid, covering 90°N to 90°S and 180°E to 180°W, along the period 1960-2022, specifically at 0 LT and 12 LT, with
the following inputs: (1) MgII as the solar activity proxy, (2) CCIR maps for foF2, (3) storm model off, (4) AMTB-2013 model
for hmF2. Considering just one day in the month or assessing the monthly median from all its daily values should give similar
results due to IRI model presents a smooth variation at daily timescale. Therefore, we considered foF2 and hmF2 values for
the 15th day of each month as equivalent to the monthly median. Selecting other days, or estimating all daily values within a
month to assess the true median, does not significantly affect the final results, as is discussed later in the Discussion Section.
A total of 37×37=1369 series were obtained for foF2 and for hmF2. In each case, annual mean series were constructed, together
with series for each of the 12 months (that is 13 series per grid point and per local time considered), all covering the period
1960-2022, which implies 63 points per series.
The foF2 and hmF2 filtering was made in the usual way estimating the residuals from a linear regression with MgII as the
solar EUV proxy (Lastovicka, 2021b, 2021c), according to:
$X_{residual} = X_{IRI} - (A * MgII + B)$,                                                   (1)
where $X_{IRI}$ is the IRI modeled foF2 or hmF2 data, and A and B are the least square parameters of the linear regression between
$X_{IRI}$ and MgII. The linear trend was assessed from the linear regression between these residuals and time, that is
$X_{residual} = \alpha t + \beta$,                                                   (2)
where t is in years and α is the desired trend in [MHz/year] for foF2, or [km/year] for hmF2. We will then have one α value
for each grid point for the annual and for the 12 monthly series. Global means were also calculated in each case using a cosine
(latitude) weighting.
The selection of MgII as the solar proxy input for IRI-Plas, and to filter foF2 and hmF2 variability linked to solar activity, is
based on recent studies which recommend the use of this index as a solar proxy for foF2 trend estimations (Lastovicka 2021b,
2021c; de Haro Barbas et al., 2021). We assume it is also the most adequate in the case of hmF2.
The MgII index was obtained from the University of Bremen. It is freely available at http://www.iup.uni-
bremen.de/UVSAT/Datasets/MgII (Viereck et al., 2010; Snow et al., 2014). The extended time series was considered in order
to cover the period previous to 1978.
To determine trends induced by Earth's magnetic field secular variation only, we also run IRI-Plas for fixed solar activity
conditions by keeping Rz constant at a mean level, while running the years from 1960 to 2022. Trends were assessed directly
through Eq. (2). A previous filtering is not needed since the only foF2 and hmF2 time variations generated by the model are
those linked to the slow changes of the modified dip at each location.
**4 Methodology to evaluate the agreement between trends based on IRI and true experimental trends**
Only foF2 was considered in the comparison between IRI and experimental trend values. In order to assess the level of
agreement between model and data, 9 stations were chosen, which are listed in Table 1. Trends were evaluated using Eq. (1)
to filter the solar activity effect and Eq. (2) to estimate trends in two ways: using the monthly median data, which will be called
experimental trends ($\alpha_{exp}$), and the IRI-Plas model output, which will be called IRI trends ($\alpha_{IRI}$).

The following metrics commonly used in data-model prediction comparisons (Willmott and Matsuura, 2005; Chicco et al., 2021) were considered to compare IRI to experimental trends: the mean relative error (MRE) and the mean absolute error (MAE). Their equations are:

$$MRE = \frac{1}{n}\Sigma\frac{(\alpha_{IRI}-\alpha_{exp})}{\alpha_{exp}}, \qquad (3)$$

$$MAE = \frac{1}{n}\Sigma|\alpha_{IRI} - \alpha_{exp}|, \qquad (4)$$

These parameters were assessed to determine overall IRI performance and also for each station separately. In the first case, summation is carried over the 9 stations considering the annual mean series, for 12 and 0 LT. In the second, summation is carried out for each station over the 12 months.

Table 1: Geographic coordinates and geomagnetic latitude of the 9 ionospheric stations analyzed to determine IRI foF2 trends accuracy.

| Station | Geographic Latitude [°] | Geographic Longitude [°] | Geomagnetic Latitude [°] |
|---|---|---|---|
| Okinawa | 26.31 | 127.59 | 17.28 |
| Wakkanai | 45.25 | 141.40 | 37.06 |
| Kokubunji | 35.71 | 139.49 | 27.28 |
| Canberra | -35.17 | 149.08 | -41.74 |
| Townsville | -19.16 | 146.48 | -26.21 |
| Hobart | -42.53 | 147.19 | -49.22 |
| Juliusruh | 54.60 | 13.40 | 53.98 |
| Boulder | 40.13 | -105.23 | 47.65 |
| Rome | 41.54 | 12.29 | 41.49 |

MRE measures the average bias of IRI trends over or underestimating the experimental trends depending on its sign: positive or negative, respectively. It gives similar information to the percentage bias, and its optimal value is 0. MAE is a scale-dependent measure of deviation that corresponds to IRI trends deviation from experimental ones. The optimal value of MAE is 0, indicating that both trends are identical.

Monthly median foF2 data from the ionospheric stations were obtained as follows. Japanese and Australian stations data are available from the National Institute of Information and Communications Technology, Japan (https://wdc.nict.go.jp/IONO/index_E.html) and the World Data Centre (WDC) for Space Weather, Australia (https://downloads.sws.bom.gov.au/wdc/iondata/au/), respectively. Both databases contain monthly medians updated to 2022. European stations monthly medians up to 2009 were obtained from Damboldt and Suessman database (Damboldt and

Suessman, 2012) (https://downloads.sws.bom.gov.au/wdc/iondata/medians/). In the case of Juliusruh, the period was updated
until 2022 with monthly medians available from https://www.ionosonde.iap-kborn.de/mon_fof2.htm.  In the case of Boulder
and Rome the period was updated with data from Lowell GIRO Data Center (LGDC) (Reinisch and Galkin, 2011). foF2 from
the Digital Ionogram Data Base (DIDBase) at LGDC has a frequency of 5 minutes. In order to obtain the monthly medians,
we first selected data with Autoscaling Confidence Score (CS) greater than 70%, and then estimated for each month the hourly
median. In the case of these two stations, it was checked that the last two years available from Damboldt and Suessman
database had a reasonable coincidence (within 95%) with the data obtained from the other two sources.

## 5. Results

### 5.1 foF2 and hmF2 trends based on IRI model, and spatial variation pattern

Fig. 1 shows foF2 and hmF2 trend values for 12 LT and 0 LT. The geomagnetic equator is also plotted for years 1960 and
2022. foF2 trends are plotted in %, which were estimated by dividing α into foF2 mean along the complete period (1960-2022)
at each grid point. In addition to overall negative trends in all cases, it can be noticed that the strongest trends occur in the
region of the greatest geomagnetic equator displacement.
The global mean trends in each case are listed in Table 2, together with the mean values of F2-region parameters, to which the
peak electron density, NmF2, was added in order to make some comparisons with other published results in the next Section.
Trends are listed in absolute and percentage values. The squared correlation coefficient, $r^2$, of each parameter and MgII is also
listed to indicate the quality of the fit to each regression model given by Eq. (1).
Table 2: Global mean values, using a cosine (latitude) weighting, of: F2-region ionospheric parameters, squared correlation
coefficient (r2) of each parameter and MgII, linear trends of filtered parameters indicated in units per decade, and the same
trends in percentage per decade.

|  | Mean | $r^2$ | α | α [%/decade] |
|---|---|---|---|---|
| foF2(12 LT) | 7.78 MHz | 0.967 | -0.10 MHz/decade | -1.31 |
| foF2(0 LT) | 4.87 MHz | 0.962 | -0.08 MHz/decade | -1.62 |
| NmF2(12 LT) | $8.05 \times 10^5$ cm$^{-3}$ | 0.970 | $-2.03 \times 10^4$ cm$^{-3}$/decade | -2.57 |
| NmF2(0 LT) | $3.18 \times 10^5$ cm$^{-3}$ | 0.963 | $-1.15 \times 10^4$ cm$^{-3}$/decade | -3.17 |
| hmF2(12 LT) | 303.1 km | 0.959 | -2.16 km/decade | -0.72 |
| hmF2(0 LT) | 323.0 km | 0.971 | -1.50 km/decade | -0.47 |

Trends assessed for each month have a similar spatial pattern as the annual trends shown in Fig. 1, even though they are not
identical. Fig. 2 (left panel) shows the global mean trends values from January to December. Weaker global trends are noticed
in February and in June. Something to notice is the decrease of $r^2$ of the fit to filter solar activity, shown in Fig. 2 (right panel).
All values are lower than the annual case. This would be due to the variation of foF2 and hmF2 associated with solar activity
being efficiently described by the 12-month moving average of a solar proxy. When analyzing the time series corresponding
to each month separately, considering the unsmoothed monthly values lowers $r^2$ because the inter-monthly variation is not
eliminated. As an additional comment, in general, when considering solar EUV proxies they are all more alike when the time
series compared consist of annual means, rather than monthly or daily means. This is because at these shorter timescale, each
time series conserves distinct variability patterns that are erased when annual or 12-month running means are used.
Fig. 3 shows trends for IRI-Plas run keeping Rz constant at a mean level (=70). These trends result then from the Earth's
magnetic field secular variation only since they reflect the modified dip changes at each location. From a comparison with Fig.
2 trend values and spatial patterns, two things become clear: (1) the positive and negative trend spatial configuration is due to
the magnetic field variation, and (2) the overall negative trends, away from the region with the pronounced geomagnetic field
equator displacement along the period considered, are not due to the magnetic field effect. Global mean trends in the case of
Fig. 3 are -0.0004 Mhz/decade and -0.086 km/decade. In percentage they become -0.0006 and -0.023 %/decade, respectively.
Comparing these values with those listed in Table 2 for 12 LT, it could be said that the global mean trend driven by the Earth's
magnetic field, despite being relatively strong at some regions, averages essentially to zero. foF2 and hmF2 global means in
this case are a 7.93 MHz and 308.6 km, similar to the Table 2 values.
**5.2 Agreement between IRI and experimental trends for selected stations**
Figs. 4 and 5 show experimental and IRI foF2 trends for each of the 9 stations, at 12 and 0 LT, respectively, in terms of months.
Error bars are estimated as one standard deviation. Generally good agreement can be noticed, which is evinced by MAE and
MRE values listed in Table 3, in particular for the 12 LT case. Annual experimental and IRI trends are listed in Table 4.
The cases with large MRE values correspond to those stations and LT that have an experimental trend value very close to zero.
Since this value appears in the denominator of MRE (see Eq. 3), even a small difference in the numerator leads to a big MRE.
However, we can re-estimate MRE's excluding experimental trends equal to zero within the error. Specifically, in the 12 LT
case, these would correspond to experimental trend values for Boulder in May; and in the 0 LT case, to Kokubunji in February
and December, Townsville in June, Juliusruh in February, and Boulder in September and October. By doing so, the MRE
decreases, as indicated by the values presented within brackets in Table 4.

Table 3: foF2 trends assessed with experimental data and with IRI-Plas model, considering annual mean data series at 12 and
0 LT. The last rows present the MAE and MRE between these trends carried over the 9 stations. MRE* corresponds to MRE
without the stations of highest relative error, that is Okinawa in the 12 LT case and Boulder in the 0 LT case.

| Station | α [MHz/decade], 12 LT | | α [MHz/decade], 0 LT | |
|---|---|---|---|---|
| | Experimental | IRI | Experimental | IRI |

| | | | | |
|---|---|---|---|---|
| Okinawa | -0.30 ± 0.07 | -0.14 ± 0.03 | -0.18 ± 0.07 | -0.18 ± 0.03 |
| Wakkanai | -0.18 ± 0.05 | -0.12 ± 0.03 | -0.04 ± 0.02 | -0.06 ± 0.01 |
| Kokubunji | -0.20 ± 0.03 | -0.13 ± 0.03 | -0.07 ± 0.02 | -0.09 ± 0.02 |
| Canberra | -0.12 ± 0.04 | -0.11 ± 0.02 | -0.05 ± 0.03 | -0.07 ± 0.02 |
| Townsville | -0.16 ± 0.05 | -0.13 ± 0.03 | -0.09 ± 0.06 | -0.10 ± 0.02 |
| Hobart | -0.13 ± 0.04 | -0.10 ± 0.02 | -0.05 ± 0.03 | -0.07 ± 0.02 |
| Juliusruh | -0.11 ± 0.03 | -0.11 ± 0.02 | -0.06 ± 0.03 | -0.06 ± 0.01 |
| Boulder | -0.08 ± 0.06 | -0.09 ± 0.03 | 0.01 ± 0.02 | -0.04 ± 0.01 |
| Rome | -0.13 ± 0.04 | -0.12 ± 0.03 | -0.10 ± 0.01 | -0.08 ± 0.02 |
| MAE | 0.04 MHz/decade | | 0.03 MHz/decade | |
| MRE | -0.19 (-19%) | | -0.79 (-79%) | |
| MRE* | -0.15 (-15%) | | -0.16 (-16%) | |

Table 4: MAE and MRE of foF2 trends assessed with experimental data and with IRI-Plas model, considering monthly data series at 12 and 0 LT, for each station. MRE values between brackets correspond to estimation excluding experimental trends equal to zero within the error.

| Station | α, 12 LT | | α, 0 LT | |
|---|---|---|---|---|
| | MAE [MHz/decade] | MRE | MAE [MHz/decade] | MRE |
| Okinawa | 0.10 | -0.39 | 0.05 | 0.27 |
| Wakkanai | 0.08 | -0.35 | 0.02 | -0.52 |
| Kokubunji | 0.06 | -0.29 | 0.03 | 1.19 (0.28) |
| Canberra | 0.02 | -0.11 | 0.04 | -0.48 |
| Townsville | 0.04 | -0.10 | 0.04 | -1.67 (-0.39) |
| Hobart | 0.04 | -0.23 | 0.02 | 0.50 |
| Juliusruh | 0.03 | 0.04 | 0.03 | 0.48 (-0.21) |
| Boulder | 0.03 | 1.07 (0.51) | 0.04 | 6.26 (0.21) |
| Rome | 0.03 | -0.08 | 0.04 | 0.26 |

## 6. Comparison with a general circulation model

The Whole Atmosphere Community Climate Model eXtension (WACCM-X) has been run to assess trends in the upper atmosphere (Solomon et al., 2018; Cnossen, 2020) and some results can be analyzed comparatively with the mean global trends here obtained with IRI-Plas, as well as the spatial variation of the trends.

WACCM-X is a general circulation and complex model with high-resolution modeling capabilities, which incorporates a comprehensive set of physics processes to estimate a more realistic representation of the atmospheric (and ionospheric) status, including chemical, dynamical, and radiative processes. This model is coupled with several Earth systems, making it easy to analyze the weight of any change in trends, e.g. the increase of a particular component in atmospheric composition. The trend results obtained by Solomon et al. (2018) and by Cnossen (2020) with WACCM-X that can be compared with those of IRI-Plas are listed in Table 5.

In the case of Solomon et al. (2018), global mean values are presented considering only minimum solar activity level and solar quiet conditions, with which no filtering is needed before the trend assessment. The period considered is 1972-2005, and there is no local time consideration, so we will assume that their values could be compared to the mean of our 12 and 0 LT values. Their trends are weaker than assessed with IRI-Plas, even if we reassess trends considering 1972-2005 instead of 1960-2022. In both cases, trends are negative, but the NmF2 trend they obtain is around half the IRI-Plas trend, as can be deduced from Table 5. In addition to trend values, NmF2 and hmF2 mean global values estimated by Solomon et al. (2018) can be compared to IRI-Plas output averages considering only years around solar minimum activity levels out of the 1960-2022 period. In this case, the results are similar for NmF2, but for hmF2 their mean value is lower than that obtained with IRI-Plas.

Cnossen (2020) presents the global mean values, as in the previous case, and the spatial pattern variation. Our trend estimation methodology is similar to Model 1 in this work, with two differences: F10.7 is used instead of MgII as solar proxy and the trend term is included in a multiple regression together with the solar activity term. The differences due to methodologies is not expected to be significant (Lastovicka et al., 2006). Absolute values of trends in this case are slightly higher than in Solomon et al. (2018), but again lower than those of IRI-Plas, with the greatest difference in the NmF2 trend case, as can be noticed from Table 5. The squared correlation coefficient, that indicates the quality of the fit to each regression model given by Eq. (1), is similar in all the cases.

Is important to remark that the trends reported by Solomon et al. (2018) may have resulted in lower values because they run the simulation with constant low solar activity. This would have neglected part of the trend that may be induced by the solar EUV flux negative trend along the last minima periods, and which we consider partly responsible for the overall negative trends observed in measured ionospheric data.

Table 5: Comparison between IRI-Plas and WACCM-X results from Solomon et al. (2018) and Cnossen (2020). All values correspond to global means along the period analyzed in each case, with the exception of NmF2 and hmF2 mean values which correspond to global mean along solar minimum activity level periods.

|  | IRI-Plas (12 LT) | IRI-Plas (0 LT) | Solomon et al. (2018) | Cnossen (2020) |
|---|---|---|---|---|
| NmF2 trend [%/decade] | -2.6±0.8 | -3±2 | -1.2 | -1.6±0.3 |
| hmF2 trend [km/decade] | -2±1 | -1.5±0.5 | -1.3 km | -1.5±0.1 |
| $r^2$(NmF2,MgII) | 0.97 | 0.96 | -- | 0.95 |
| $r^2$(hmF2,MgII) | 0.96 | 0.97 | -- | 0.94 |
| NmF2 mean [cm$^{-3}$] | $2.14 \times 10^5$ | $1.39 \times 10^5$ | $1.74 \times 10^5$ | -- |
| hmF2 mean [km] | 302.3 | 269.5 | 259.6 | -- |



The spatial variation pattern can also be compared. In the case of hmF2, Cnossen (2020) spatial pattern is consistent with IRI-
Plas trends at 12 LT, with overall negative trends and a positive patch above the geomagnetic equator between Africa and
South America. This would be in agreement with the trend expected from the northward geomagnetic equator secular
displacement, which is strongest in this region, and assuming that the equatorial ionization anomaly (EIA) pattern of hmF2
moves along with this displacement. The highest decreasing trends are observed consequently below this positive patch, in
response to the northward movement of hmF2 highest values. With respect to trend values, the strongest positive trends seem
similar, around 2 km/decade. However, the highest negative trends are greatest in IRI-Plas case, reaching values of 12
km/decade at noontime while in Cnossen (2020) case this value corresponds to 5 km/decade.
In the case of hmF2 trends at 0 LT, hmF2 presents a trough, even though not as well defined as the crest during daytime hours.
The displacement of this trough attached to the geomagnetic field northward displacement induces an effect inverse to that
during noon. That is a positive trend patch appears below it, with the strongest negative trends above.
A similar situation occurs with foF2 trend spatial pattern. In order to compare the trend values in percent with those of NmF2,
they should be multiplied by two. This means that our strongest negative trend is again the highest. The spatial pattern here
has alternating bands of positive and negative trend values aligned with the EIA, which can be explained in terms of the EIA
displacement following the geomagnetic equator. Between ~60°W and 0° in longitude the equator shift is the greatest and
northward, so this is the region where the strongest alternating trends are noticed (Elias et al., 2022). This longitudinal extension
is narrower than in Cnossen (2020) case, who detect it between ~60°W and ~20°E. A notorious difference is that between the
initial and the final position of the geomagnetic equator in this longitudinal range, Cnossen (2020) detects a negative trend
band while in our case a positive band is observed.
This difference may be caused by a poor resolution in latitude. In order to see the trend bands expected in the region between
the initial and final position of the equator, a schematic plot is shown in Fig. 5(a) of the EIA foF2 trough in its initial and its
final position in 1960 and 2022, respectively. In this figure, it can be clearly noticed that the region between these positions
will present a positive portion followed by a negative one. The first one corresponds to the region with low foF2 in 1960,
which has now become a region with higher foF2 values (since the trough has moved). The second one corresponds to a region
of higher foF2 in 1960, which now is located under the EIA trough. On average, the geomagnetic equator has displaced ~5 to
10° in the region's strongest shift, so for low resolutions, the grid points may coincide with one of either trend bands. This
could partly explain the difference between Cnossen (2020) negative band between the equator positions, and our
corresponding positive band. Fig. 5(b) shows an enlarged portion of the trends spatial pattern obtained with IRI-Plas, but
increasing the latitude resolution to 1°, where it can be noticed a positive and a negative band within the limits of the 1960 and
the 2022 equator positions.
The spatial pattern linked to the EIA displacement following the geomagnetic equator and clearly isolated in Fig. 3, is expected
in IRI-Plas foF2 and hmF2 modeling since the model includes a real geomagnetic field. Even though there are very few stations
along its location, IRI model reproduces the EIA pattern through the variation in the magnetic inclination, obtained from IGRF,
on which interpolation coefficients depend.

### 7. Discussion

It is worth noting that in very recent works, the 30 cm solar flux index, F30 (available at
https://spaceweather.cls.fr/services/radioflux/), is identified as the most suitable EUV solar proxy for filtering foF2 to
subsequently estimate long-term trends, followed by MgII (Lastovicka and Buresova, 2023). We also conducted a recent study
(Zossi et al., 2023) where we concluded that both F30 and MgII are equally appropriate, but without being able to distinguish
which one is better of the two. In the present work, we did not consider F30 because IRI-Plas does not have this option.
However, we compared the trend values for the 9 stations here analyzed, from measurements and IRI-Plas model, considering
each of these indices to filter solar activity effect, and while we did not obtain identical values, they are in strong agreement.
This agreement is nearly complete in terms of sign, and practically within the error range of the trends in terms of values.
Nevertheless, this deserves a detailed comparative analysis, and could possibly suggest the inclusion of F30 as an additional
index to the options already available in this model.
Another important aspect concerns IG and Rz long-term trends. The explained variance of each of these proxies by MgII is
~95% ($r^2 \times 100$) in both cases (IG vs. MgII and Rz vs. MgII) and, if the solar activity effect is filtered from them through the
same linear regression as that performed on F2 parameters, negative trends are obtained in both residuals as is shown in Fig.
7. The decreasing trend observed in IG when filtered with MgII, is mainly due to the last two solar cycle minima which are
much lower than the previous two in IG case (and also in Rz case) than in MgII case. This is due to the solar EUV flux during
the last two minima has been lower than the values indicated by solar proxies. This would also induce a decreasing trend in
foF2 (and in hmF2) which might be connected to the inadequate performance of the proxy in capturing the variations in solar
EUV flux (Emmert et al., 2010; Chen et al., 2011; Bruevich and Bruevich, 2019; Elias et al., 2023). However, this is a topic
for further research.
Other aspect to discuss is the use of foF2 and hmF2 values for a single day of each month as representative of the monthly
median. On one hand, as mentioned in Section 3, the IRI model, with the "storm off" option, exhibits a smooth daily variation

throughout each month. To analyze the impact of choosing this particular day on the trend instead of the monthly median or mean obtained from all daily values, we assessed annual noon foF2 trends for a mid-latitude location (20°N, 30°E) considering other days of each month (but using the same day for every month and year). We also assessed the trends by considering the median and the mean value of each month. Even though the trend values are not the same, the difference between any of them is around ~0.006 Mhz/decade that is smaller than the trends' standard error (~0.02 MHz/decade). As an additional possibility, we considered using a random day in each month. For example, for year 1960: day 12 for January, day 27 for February, day 5 for March, and so on for the following months and years. From 10,000 random estimations we made, the minimum trend value obtained is -.09 MHz/decade, and the maximum value is -.13 MHz/decade. Both include within the error interval (±0.02 MHz/decade) the value of the trend obtained considering day 15 (which is -0.0110 MHz/decade), and that considering the true foF2 median (which is -0.0111 MHz/decade). The most probable trend values in this running of 10 thousand trend estimations lies between -0.111 and -0.109, and it again includes the value estimated in this work considering day 15.

As an additional topic deserving further research is the global picture easily obtained with IRI of the geomagnetic field secular variation induced trends. We will not go deeper into this aspect in this work, but we considered it important to mention that the positive and negative trend patches are consistent with the results by Cnossen and Richmond (2012), who analyzed the effect of the Earth's dipole inclination variation using the Coupled Magnetosphere-Ionosphere-Thermosphere (CMIT) model. The only difference is that they show the effect of a dipole axis increasing its inclination, and the secular variation observed during the last decades is consistent with a dipole aligning with the rotation axis. That is why, in a rough comparison, the sign of the trend's patches in Fig. 3 are opposite to those of Fig. 7 (lower panels) of Cnossen and Richmond (2012). This is something worth exploring, using IRI, at least as long the field remains mainly dipolar.

**8. Conclusions**

Considering how the foF2 and hmF2 interannual variation is determined in IRI-Plas, and in other IRI model versions, it can be argued that the overall negative trends are due to the same long-term trend occurring in IG and Rz.

For foF2 the attribution to external forcings other than the magnetic field is clear since IG carries the information of foF2 measurements. Thus, we can expect that the trends obtained could be a reasonable approach to experimental trends. In fact, this index includes the variability by other sources affecting the ionospheric F2 layer, like the greenhouse gas concentration increases or other neutral composition changes or dynamical disturbances. It does not include, however, the magnetic field secular variation effect since it averages to almost zero. Hence, the foF2 trends obtained using IRI-Plas model values can be, to a first approximation, attributed to the greenhouse cooling effect plus the secular variation in Earth's magnetic field. This, of course, assumes that the dominant driver behind the global declining foF2 trend, and of IG, is indeed the greenhouse effect. In addition, to verify this in a more localized spatial scale, we were able to compare foF2 trends, considering annual and monthly series, determined with experimental data from 9 mid-latitude stations and with the corresponding IRI-Plas modeled

values. We obtained a reasonable agreement, with average differences of ~20%. Something to argue is that by using mid-latitude stations, we use the stations for which IRI surely works best.

On the other hand, hmF2 trends result from the Rz overall decreasing trend when it is filtered with MgII (or any other solar EUV proxy). Even though the values are coherent with expected trends due to greenhouse cooling (due to Rz varies almost identically to IG) we cannot conclude, using IRI, that its long-term lowering is due to greenhouse gas concentration increases. This is due to the coincidence that both hmF2 from IRI, and hmF2 from measurements and theoretical considerations, are forced by a "mechanism" inducing a downward trend: in the first case, it is the Rz overall downward trend along the period considered, while in the latter it would be due to the greenhouse effect. In addition, of course, the downward trend in Rz has nothing to do with the increasing concentration of greenhouse gases during the last decades. They both just happen to be in the same direction. Despite this, it is considered worthwhile to compare these trends with experimental values as a future task. Smaller Rz values since ~2001 have been mentioned some years ago by Lukianova and Mursula (2011) and Mielich and Bremer (2013).

To be able to attribute observed trends in IRI to processes unrelated to solar activity, it would be valuable to consider two potential approaches:

1. Interpolation of the CCIR maps with "effective indices" derived from data, as already proposed by Pignalberi et al. (2018). This would be similar to the approach used for IG. By using effective indices based on observed data, time variations not directly tied to solar activity would be accounted for.

2. Interpolation from annual CCIR maps, instead of the two maps currently used. This would involve updating the CCIR maps on an annual basis assimilating the most recent and accurate data, and thus the time variation obtained would not result exclusively from solar activity variability.

Compared to a more theoretically based model, it is important to remark that IRI-Plas is designed for modeling a specific atmospheric sub-region, namely the ionosphere, whereas WACCM-X is a global circulation model that simulates the entire atmosphere. Thus, an advantage of this model is that, in considering coupling processes among several Earth systems, it allows to analyze the weight of any change in trends, e.g. the increase of a particular component in atmospheric composition. Nevertheless, the negative side of general circulation models is the substantial computational resources and time needed to run simulations, being almost exclusive for high-performance computing centers. In the case of IRI a great advantage is its user-friendly design, allowing it to be run on modest computers consuming few resources with extremely short computational times (in the order of minutes), while still being a reliable tool to get an approximated status of the ionosphere. On the other hand, this reference model adopts simplified assumptions and parameterizations to represent complex ionospheric processes.

Before summarizing the answer to the question raised by this work's title, we bring up again some recommendations for future tasks suggested throughout this work: (1) a comparison between hmF2 trends at specific locations between IRI and ionosonde data, similar to foF2 analysis, (2) the spatial pattern of IRI trends due only to the Earth's magnetic field and its comparison with complex models with theoretical approaches, (3) the correct attribution of the general foF2 and hmF2 downward trend: the greenhouse affect or a long term decreasing EUV flux not shown by EUV proxies which are used to filter solar activity

effect since Rz was suggested to be discarded for this purpose (Mielich and Bremer, 2013), (4) modify IRI effective proxies
or coefficient maps in order to determine foF2 and hmF2 interannual variations that include external sources other than solar
activity only.
In summary, regarding the question set forth in this study's title, we conclude that the IRI model can be a valuable tool for
obtaining preliminary approximations of experimental trends, at least in the case of foF2. This is particularly significant given
the low spatial density of data and the scarcity of series with sufficient length to estimate trends. In the case of hmF2, there
would be an added advantage considering that, while foF2 is an accurately measured parameter, hmF2 is often missing or
derived from the proxy M(3000)F2 parameter. However, even if hmF2 trends obtained with IRI-Plas are close to the expected
values, they are linked to different drivers.

## Code and Data availability

The IRI-Plas code is freely available at www.ionolab.org/ and from the IZMIRAN Ionospheric Weather server at
https://www.izmiran.ru/ionosphere/weather/grif/SPIM/. The IRI 2016 and 2020 versions, also used in this work, was run from
the HF propagation toolbox, PHaRLAP, created by Dr. Manuel Cervera, Defence Science and Technology Group, Australia
(manuel.cervera@dst.defence.gov.au), available at https://www.dst.defence.gov.au/our-technologies/pharlap-provision-high-
frequency-raytracing-laboratory-propagation-studies. This toolbox is available by request from its author. foF2 data from
Japanese and Australian stations are available at https://wdc.nict.go.jp/IONO/index_E.html, and
https://downloads.sws.bom.gov.au/wdc/iondata/au/), respectively. Boulder and Rome foF2 were obtained from
https://downloads.sws.bom.gov.au/wdc/iondata/medians/ and http://spase.info/SMWG/Observatory/GIRO. Juliusruh foF2 is
available from https://www.ionosonde.iap-kborn.de/mon_fof2.htm. MgII index is freely available at http://www.iup.uni-
bremen.de/UVSAT/Datasets/MgII, IG from https://www.ukssdc.ac.uk/cgi-bin/wdcc1/secure/geophysical_parameters.pl, Rz
from https://www.sidc.be/SILSO/datafiles.

## Author contribution

B.S. Zossi: conceptualization, supervision, investigation, formal analysis, methodology, and writing; T. Duran: investigation,
methodology, validation, review and editing; F.D. Medina, B.F. de Haro Barbas and Y. Melendi: investigation, validation and
review; A.G. Elias: original draft preparation, investigation, formal analysis, review and editing.

## Competing interests

The contact author has declared that none of the authors has any competing interests.

## Acknowledgements

B.S. Zossi, F.D. Medina, B.F. de Haro Barbas and A.G. Elias acknowledge research project PIP 2957. T. Duran acknowledge research projects PICT 2019-03491 and PGI 24/F083. We are very grateful to David Themens for his comments on our article and for having the time for a fruitful and helpful discussion which helped greatly to improve the manuscript. We also gratefully acknowledge two anonymous reviewers for their valuable and insightful comments.

We acknowledge the International Reference Ionosphere (IRI) working group and the Defence Science and Technology Group (DST) of Department of Defence, Commonwealth of Australia for providing the HF propagation toolbox, PHaRLAP, created by Dr. Manuel Cervera, from which IRI 2016 and 2020 were run. We acknowledge IONOLAB (www.ionolab.org/) and IZMIRAN Ionospheric Weather server (https://www.izmiran.ru/services/iweather/) for providing IRI Plas software. We also acknowledge GIRO data resources http://spase.info/SMWG/Observatory/GIRO, the WDC for Ionosphere and Space Weather, Tokyo, National Institute of Information and Communications Technology, the Australian Space Weather Forecasting Centre (ASWFC), and the Leibniz Institute of Atmospheric Physics for providing foF2 data.

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

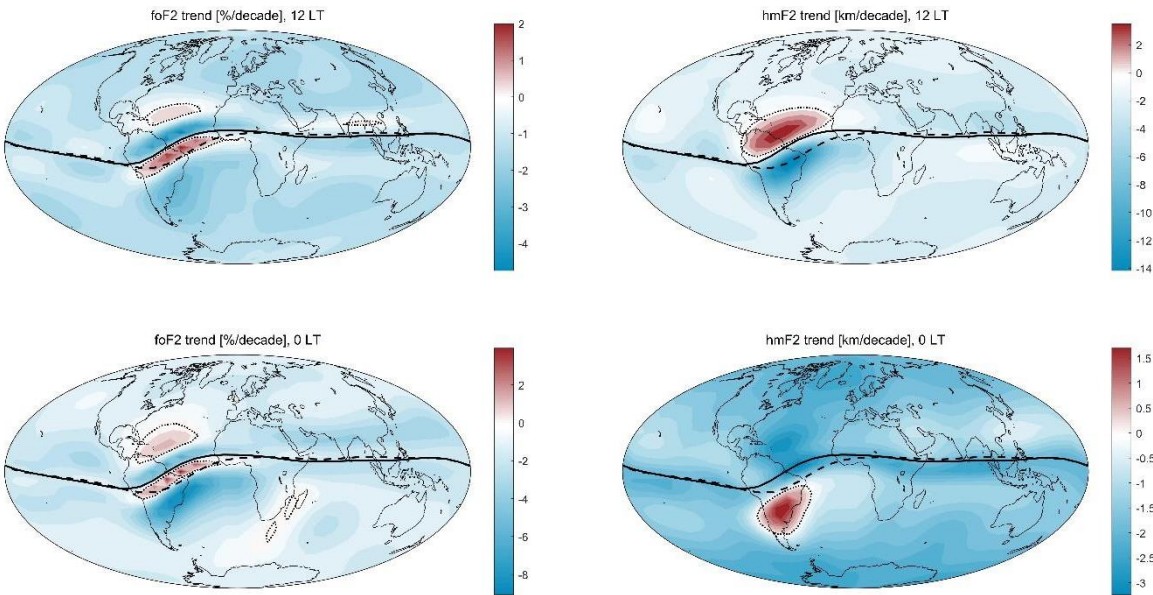


**Figure 1: Trends of foF2 (left panels) and hmF2 (right panels), at 12 LT (upper panels) and 0 LT (lower panels) along the period 1960-2022 assessed with IRI outputs, which were previously filtered using Eq. (1). Note: Trends are indicated per decade, and foF2 trends are in percent. Enhanced dashed and solid lines indicate the magnetic equator position in 1960 and in 2022, respectively. Dotted lines indicate zero trend.**


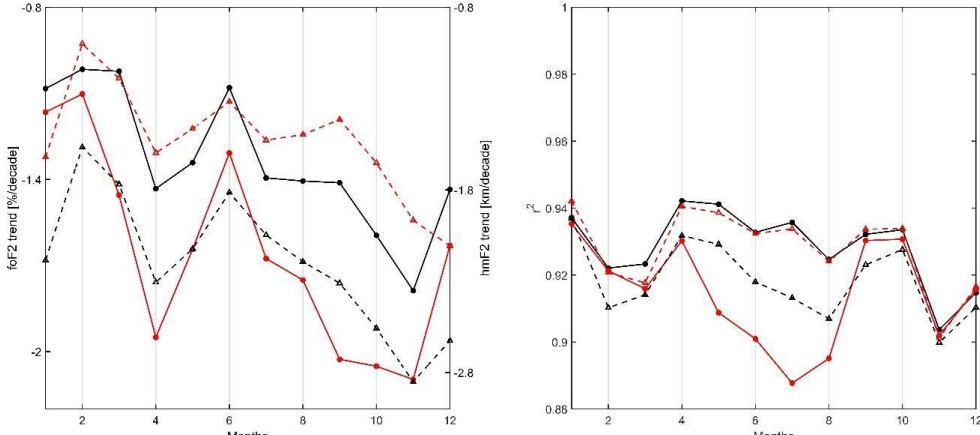


**Figure 2: Global mean values of the linear trends (left panel) and the squared correlation coefficient (r2) between each parameter and MgII (right panel) of foF2 (solid line with circles) and hmF2 (dashed lines with triangles) at 12 LT (in black) and 0 LT (in red).**


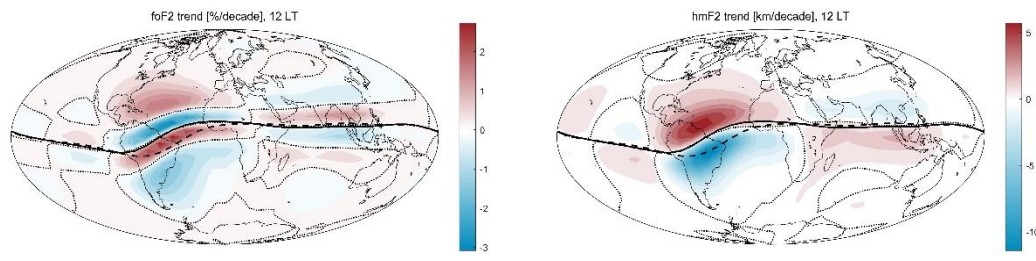


**Figure 3: Trends of foF2 (left panel) and hmF2 (right panel), at 12 LT along the period 1960-2022 assessed with IRI outputs using Eq. (1), without previously filtering. Note: Trends are indicated per decade, and foF2 trends are in percent. Enhanced dashed and solid lines indicate the magnetic equator position in 1960 and in 2022, respectively. Dotted lines indicate zero trend.**


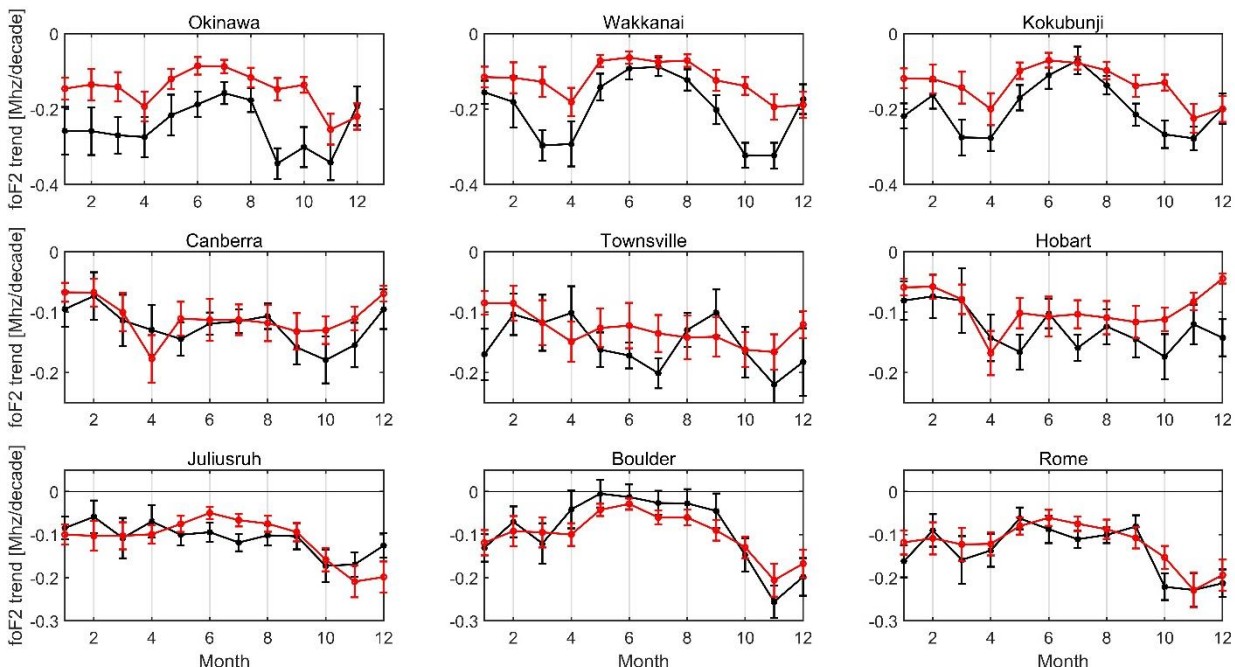


**Figure 4: Monthly variation of foF2 trends in [MHz/decade], at 12 LT, estimated with experimental data (black) and with IRI-Plas model (red). Error bars correspond to one standard deviation.**


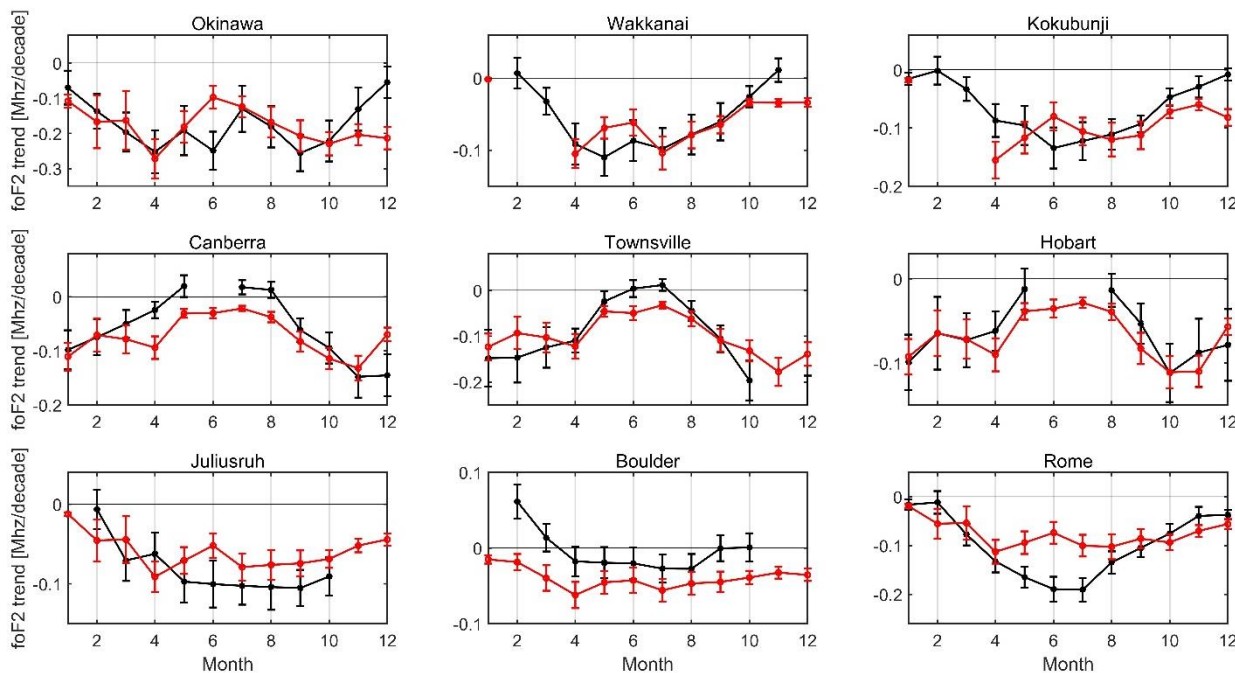

**Figure 5: Monthly variation of foF2 trends in [MHz/decade], at 0 LT, estimated with experimental data (black) and with IRI-Plas model (red). Error bars correspond to one standard deviation.**

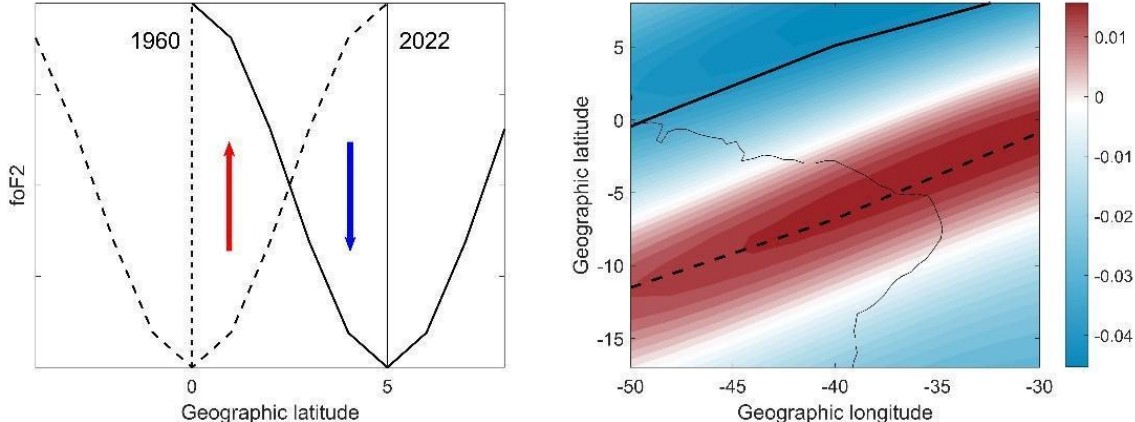

**Figure 6: (Left panel) Schematic representation of foF2 latitudinal profile around the EIA trough in 1960 (dashed line), centered in the magnetic equator in 1960 (vertical dashed line at latitude=0), and in 2022 (solid line), centered in the magnetic equator in 2022 (vertical solid line at latitude =5). The red arrow indicates the foF2 increase that would be observed in latitudes between 0 and ~2.5, and the blue arrow the decrease between ~2.5 and 5. (Right panel) foF2 trend along 1960-2022 assessed with IRI-Plas (solar activity filtering with MgII) in the region with the largest equator displacement with an increased resolution: 1°×2° latitude-longitude grid. Enhanced dashed and solid lines indicate the magnetic equator position in 1960 and in 2022, respectively.**

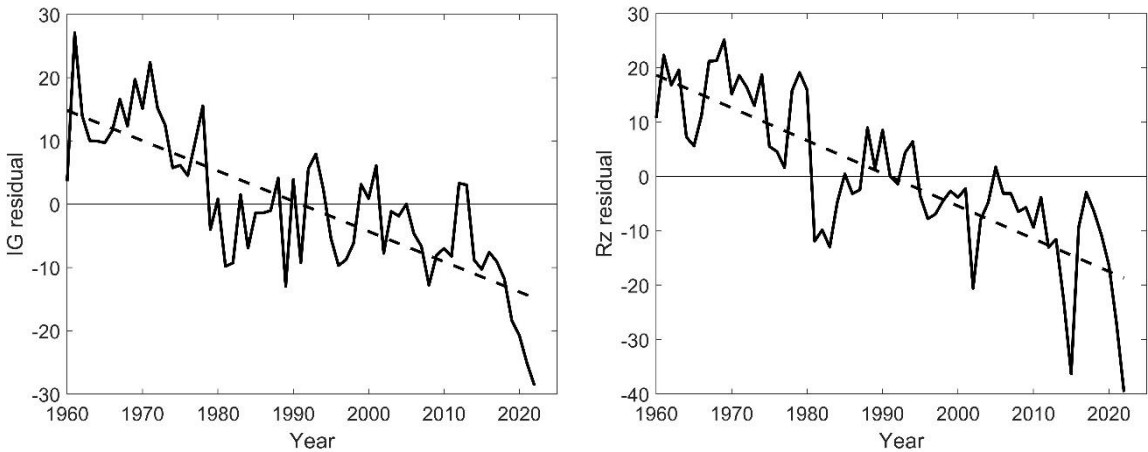

544

545 **Figure 7. Residuals of the linear regression between annual means IG and MgII (left panel), and Rz and MgII (right panel).**