# Peer review of "Evaluating F2 region long term trends using the IRI model: A feasible"

_EGUsphere, 2023_

## Author Comment (AC1)

Thank you very much for your comments and corrections.

Following are our answers (in blue) to your comments (in black).

In the revised manuscript, the changes which correspond to your remarks will appear in red, together with those corresponding to the comments of Reviewer #2 and to Dr. David Themens.

Line 46: "geomagnetic equator" should be "geomagnetic equator in some longitudinal ranges" – to be more correct.

You are right, since there some longitude ranges where the geomagnetic equator displacement is negligible. We will add: "in some longitudinal ranges", in the revised version of our manuscript.

Lines 92-93: The 15th day of a month need not be good approximation of monthly medians. If the 15th day occurs in the maximum or minimum of the 27-day variation, then it differs significantly from monthly median. Smooth variation in IRI does not mean no variation. However, we can assume that in the case of large number of data as it is the case of long-term trend investigations the effects of maxima and minima of the 27-day variation essentially cancel out. Nevertheless the usage of monthly medians in future work would increase reliability of results.

foF2 modeled by IRI does not include the 27 day variation, but an almost linear variation, which makes the mid-day of the month (around day 15) quite close to the median. As an example, Figure R1 shows the daily foF2 along one year (2000) for a point at 20°N, 30°E, 12 LT.

[Figure]

Figure R1. Daily foF2 estimated with IRI model at 20°N, 30°E, 12 LT, year=2000 (January to December). Dashed vertical lines correspond to the end of each month.

In Figure R2, we show the first three months alone, so the daily values can be noted more clearly.

[Figure]

Figure R2. Daily foF2 estimated with IRI model at 20°N, 30°E, 12 LT, year=2000 (January to March). Dashed vertical lines correspond to the end of each month.

However, you are correct in that the foF2 IRI value within a month can differ depending on the chosen day. In order to assess the effect of choosing a given day, as is done in our work in order to save computing time, and of using the median we have done the following "experiment":

We first estimate the daily foF2 values along the period 1960-2022 for a mid-latitude location (20°N, 30°E). From this series, we assessed the annual foF2 value, but instead of using day 15 to represent the foF2 median of each month, we used day 1, then day 2, and so on until day 28 (we did not consider days 29, 30 and 31 due to February's days). In this way we obtained 28 different foF2 annual series. For each of them we estimated the trend after filtering solar activity effect using MgII. The values obtained are shown in Figure R3. We also assessed foF2 trends by considering the annual foF2 value averaging the median value of each month, and the mean of each month.

[Figure]

Figure R3. foF2 trend [MHz/decade] (black dots) in terms of the day of the month used to represent the monthly mean, which was then used to estimate foF2 annual mean. Solar activity was filtered with MgII. foF2 trend estimated from annual means obtained by averaging monthly median and monthly means of each month are indicated as a blue and a red dashed line, respectively. Note that the trend standard error is 0.02 MHz/decade, that is higher then the trend difference between black dots and any of the dashed lines.

Even though the trends are not the same, neither between them nor between them and the values obtained considering medians or means, the difference is smaller than the standard error of these trends, which in all the cases is ~0.02 MHz/decade. Note that the difference is around 0.006 Mhz/decade (an order of magnitude smaller than the error).

As an additional possibility we assessed foF2 annual values averaging the 12 monthly values assessed using a random day in each month. For example, for year 1960: day 12 for January, day 27 for February, day 5 for March, and so on for the following months and years. In this last case we made 10,000 random estimations which are shown un the following histogram (Figure R4).

[Figure]

Figure R4. Histogram of 10,000 trends based on annual foF2, 12 LT, at position 20°N, 30°E, estimated considering one random day per month. Solar activity is filtered through Mg II.

The minimum trend value obtained is -0.09 MHz/decade, and the maximum value is -0.13 MHz/decade. Both include within the error interval (±0.02) the value of the trend obtained considering day 15 (which is -0.011024 MHz/decade), and that considering the true foF2 median (which is -0.011069 MHz/decade). The most probable trend values in this running of 10 thousand trend estimations lies between -0.111 and -0.109, and it again includes the value estimated in this work considering day 15.

We will add a comment on this trend variation depending on the day selection in IRI model in the Discussion section of the revised version of our manuscript.

Lines 106-108: Lastovicka and Buresova (2023) recommended F30 as the best solar proxy followed by Mg II used by authors. However, I understand that authors prepared their paper essentially before the paper by Lastovicka and Buresova has been published and Mg II appears also useful solar proxy.

F30 is not available as a solar proxy in IRI-Plas model. However, we estimated the trends with measured foF2 data at the 9 stations included in our work. They do not differ much from those estimated considering MgII, as can be seen in Figure R5 in the annual case at 12 LT, even though there are some cases differing in more than 0.2 MHz/decade (which is the average standard error for most of the trend values assessed with MgII or with F30). We also estimated these trends for the same stations but with foF2 assessed from IRI-Plas model.

[Figure]

Figure R5. foF2 trend values [MHz/decade], based on annual data at 12 LT, estimated from stations' measured data (Obs, for 'observations') and IRI-Plas assessed data for each station, filtering solar activity effect considering MgII and F30: black: observations filtered with MgII, blue: observations filtered with F30, red: IRI-Plas estimations (using MgII) filtered with MgII, green: IRI-Plas estimations (using MgII) filtered with F30.

We perform the same estimation for each month along the period 1960-2022 (equivalent to Figure 3 of our work), at 12 LT, and we still see a good agreement, as can be seen in Figures R6 and R7, for measured f0F2 and IRI-Plas foF2, respectively.

[Figure]

Figure R6. foF2 trend values [MHz/decade] for each month at 12 LT, estimated from stations' measured data filtering solar activity effect considering MgII and F30: black: observations filtered with MgII, blue: observations filtered with F30.

[Figure]

Figure R7. foF2 trend values [MHz/decade] for each month at 12 LT, estimated from IRI-Plas assessed data for each station, filtering solar activity effect considering MgII and F30: red: IRI-Plas estimations (using MgII) filtered with MgII, green: IRI-Plas estimations (using MgII) filtered with F30.

In the revised version of our work we included a comment based on these results in Section 6 (Discussion and conclusions) and added the reference of Lastovicka and Buresova (2023). In fact, we have also worked on F30 adequacy to to filter solar activity from foF2 and compared it to other indices, concluding that both, MgII and F30, are the best for this purpose. However, we could not distinguish if one was better than the other one. Our work (Zossi et al., 2023) was published after we sent this study to ACP, so we just now are including its reference.

Zossi, B.S., Medina, F.D., Tan Jun, G., Lastovicka, J., Duran, T., Fagre, M., de Haro Barbas, B.F., and Elias, A.G.: Extending the analysis on the best solar activity proxy for long-term ionospheric investigations, Proc. R. Soc. A., 479, 202302252. doi:10.1098/rspa.2023.0225

Line 161: "Stronger" should be "Weaker" according to Fig. 2 – trends in February and June are only about -1%/decade.

You are correct. We have change "stronger" fro "weaker" in the revised version of our work.

Page 8, Table 4: Some MREs, particularly for 00 LT, are too high – e.g. for Townsville α is not small at 00 LT (the second highest), nevertheless the corresponding MREs are very high. Please make a comment on that in the paper with possible explanation.

The explanation for this, and other large MRE values is given in the following paragraph, where precisely Townsville at 00 LT value is included:

"The cases with large MRE values correspond to those stations and LT that have an experimental trend value very close to zero. Since this value appears in the denominator of MRE (see Eq. 3), even a small difference in the numerator leads to a big MRE. However, we can re-estimate MRE's excluding experimental trends equal

to zero within the error. Specifically, in the 12 LT case, these would correspond to experimental trend values for Boulder in May; and in the 0 LT case, to Kokubunji in February and December, Townsville in June, Juliusruh in February, and Boulder in September and October. By doing so, the MRE decreases, as indicated by the values presented within brackets in Table 4."

In summary, the reason is the small trend values (close to zero) for these cases. So, if we compare to very close values we will anyway obtain a large MRE due to the division has a denominator close to zero.

Line 218: Delete "(highest values above the geomagnetic equator)" – this is unnecessary and incorrect statement.

Sorry for this mistake. We will delete this comment in the revised version.

Line 269: "represented by" should be "derived from" – this is more accurate.

You are correct. We will make this change.

Wording and misprints:

Line 107: "based in recent" should be "based on recent"

We will make this change.

Line 235 and throughout the paper: "valley" – the term used usually in literature is "trough"

Thank you for this observation. We will change "valley" for "trough" in the 7 places it appears along the manuscript.

Please, notice that after considering the observation made by Dr. David Themens some conclusions and arguments based in IRI model run have changed. They will be clearly stated in the revised version of our work.

Hoping to meet all your requirements,

Bruno S. Zossi, Trinidad Duran, Franco D. Medina, Blas F. de Haro Barbas, Yamila Melendi, and Ana G. Elias

---

## Author Comment (AC2)

Thank you very much for your comments and corrections.

Following are our answers (in blue) to your comments (in black).

The changes in the revised manuscript which correspond to your remarks will appear in red, together with those corresponding to the comments of Reviewer #1 and to Dr. David Themens.

The major issue that was not clear to me is the role of greenhouse gases in these trends (also raised in the comment by David Themens). The authors state at line 247 "The overall negative trends in both, foF2 and hmF2, is in agreement with that expected from increasing greenhouse concentration. Taking into account that IRI model does not include any forcing linked to these gases, the trends observed can be attributed to the data." What does this second sentence mean? What is the "data" being referred to? If the IRI model is periodically fitted to ionosonde observations, which are affected by greenhouse gas-induced changes, then it must already implicitly incorporate the effect of greenhouse gases. Although you state at line 80: "According to IRI general specifications, we expect it to somehow force variations linked to changes in the geomagnetic field, since it uses the IGRF model to specify geomagnetic poles and equator, but not those variations expected from the increasing greenhouse gases concentration." This is all very unclear.

We will explain now in more detail the sources of the trends when they are estimated considering foF2 and hmF2 obtained from IRI.

And precisely regarding your specific comment: "If the IRI model is periodically fitted to ionosonde observations, which are affected by greenhouse gas-induced changes, ...", it is not the IRI coefficients which are periodically adjusted for each year, but the solar activity proxy used, that is the IG index which carries the observations' information.

The trends expected from the secular variation of the Earth's magnetic field are clearly due to the interpolation coefficients with which foF2 and hmF2 are calculated, since they depend on the magnetic field inclination, and are obtained from IGRF. So, its secular variation is seen in foF2 and hmF2, which depends on location.

In the revised version of our work we will include the following paragraphs which explain in detail how IRI assess foF2 and hmF2:

"A key aspect in the present study is how IRI determines the F2 parameters for a given location. To begin, foF2 is obtained from CCIR (Consultative Committee on International Radio) maps that are based on a procedure of numerical mapping of a set of coefficients (CCIR Atlas of Ionospheric Characteristics, 1991) determined from a fitting to observed monthly median foF2 data from a worldwide network of ionosonde stations (~150 in total). From these maps of coefficients, IRI model reproduces the diurnal, seasonal and solar activity variation of foF2 in terms of latitude and longitude through Fourier time series. First, there is a set of functions in terms of geographic coordinates and the modified dip latitude used to describe the variation of the Fourier coefficients for a given number of harmonics defining the diurnal variation. Then, the seasonal variation is taken into account through a set of these coefficients (988 in total) for every month of the year. And finally, the solar activity dependence is considered by having all these monthly coefficients that account for the diurnal and geographic variation for two different activity levels: IG12=0 and IG12=100. From a linear fit between these two extremes (and also out of this range), the harmonic coefficients for any solar activity level can be estimated. IG

was originally computed using 13 globally distributed ionosonde stations that included two of the 9 stations here analyzed: Kokubunji and Canberra (Liu et al., 1983). The distribution of these stations was a compromise between good global coverage and reliable long operating ionosonde stations. Due to station closings and data unavailability, the number of stations used in IG has decreased to four, but still includes the two stations which are included in the present study (Brown et al., 2018). Therefore, this proxy, being obtained from ionospheric measurements, includes foF2 variations not covered by a solar index.

Specifically, when a given solar proxy is selected among the IRI-Plas 8 options, it is automatically converted to other related indices used by the different modules procedures (Gulyaeva et al., 2018). In this way, foF2 interannual variation is obtained from the IG12 of the selected date. This index value is which finally defines the CCIR maps coefficient values that are assessed, as already mentioned, from the linear interpolation between the two coefficient sets, one for IG12=0 and the other for IG12=100.

Turning to the case of hmF2, the default option is considered in this study, and corresponds to the AMTB-2013 model (standing for Altadill-Magdaleno-Torta-Blanch) (Altadill et al., 2013). This model is based on quiet ionosphere data from 26 digisondes collected between 1998 and 2006. The monthly averages of the global hmF2 variations are represented by spherical harmonics including modified dip latitude and longitude for two selected levels of Rz12 (0 and 100, as in the case of IG). The interannual variation of hmF2 is obtained then from a linear fit of these two levels considering the Rz12 value of the corresponding date. The same procedure is applied in the cases of the other two options for hmF2 modeling. Thus in hmF2 case, the proxy used is only reflecting solar activity variability. Nevertheless, we include its long-term trend analysis considering that the correlation between IG and Rz is higher than 0.99, and that for a given location and hour, foF2 and hmF2 interannual variation highly correlates. Moreover, IG correlates the highest with Rz exceeding 0.99 along the period 1960-2022. The linear correlation between IG and MgII, F10.7 or Lyman-α, for example, are 0.975, 0.985 and 0.970 respectively."

Since the IRI model is fitted to ionosonde data, it is surely to be expected that there will be good agreement with the ionosonde data shown in Figs. 3 and 4. It seems rather circular, so I don't understand what the comparison really tests. It would be very helpful to provide a deeper description for the reader of exactly how the IRI model is fitted to ionosonde data e.g. how often the fitting takes place, over how many stations, are satellite measurements also used?

Thank you for this observation, which complements that of David Themens. We will explain now the process of how IRI assesses the time variation of foF2 or hmF2 for a certain location (included in the answer to your previous comment). This makes clear that, even though this ionospheric model uses foF2 measurements, it does it through a global index which is "processed" to finally give the selected location data.

In addition, even though it can be "circular", the fact is that the stations data is very sparse compared to the whole planet. So, the utility of the model is precisely "circular" at the stations whose data was included, but it is useful for the estimation at locations where there is no measured data.

Note that the case of hmF2 is different, and our conclusions regarding hmF2 trends, even though we obtain values according to expected ones, we cannot argue that they are due to the greenhouse cooling.

Minor points and corrections

line 75: "used to fix solar..."

The idea of the sentence is that IRI uses a given solar proxy which you cannot change. Not that the model fixes the solar activity level.

We have written this idea clearer in the revised version.

line 78: "we decided to ..."

We will make this correction in the revised version of our work.

line 90: define CCIR maps

We will include now the definition and additional explanation of CCIR maps in Section 2 (On some aspects of the IRI model), together with additional explanation on how IRI model takes into account the Earth's magnetic field (as included in our answer to your first comment).

line 162: Figure 2 does not contain upper and lower panels

You are correct. They correspond to left and right panels. We will make this correction to the revised version.

line 169: "generally good agreement"

We will make this change in the revised version of our manuscript.

line 205: "in the NmF2 trend case..."

We will make this correction in the revised version of our work.

line 214: "hmF2, the Cnossen (2020)..."

We will make this correction in the revised version of our work.

line 241: "...the Cnossen (2020) negative band"

We will make this correction in the revised version of our work.

line 254: "...to the hmF2 case."

We will make this correction in the revised version of our work.

Please, notice that after considering the observation made by Dr. David Themens some conclusions and arguments based in IRI model run have changed. They will be clearly stated in the revised version of our work.

Hoping to meet all your requirements,

Bruno S. Zossi, Trinidad Duran, Franco D. Medina, Blas F. de Haro Barbas, Yamila Melendi, and Ana G. Elias

---

## Author Comment (AC3)

David, thank you very much for your comments and the time you took to explain us essential aspects of IRI model.

Following are our answers (in blue) to your comments (in black).

The changes in the revised manuscript which correspond to your remarks will appear in red, together with those corresponding to the comments of Reviewer #1 and Reviewer #2.

Realistically, longterm trend in the IRI can only be attributed to processes that adhere to changes in the drivers of the model themselves. As the IRI does not include a greenhouse gas-related index or driver and does not include any longterm trend parameters except solar activity, it cannot represent the impacts of that in its output. There is no multi-year term in the IRI parameterization, except solar activity, that would allow it to represent such trends even if they existed in the data used to fit the model. The IRI is just an interpolation between a low solar activity and a high solar activity map of foF2 and M3000F2, it doesn't care about the year or date outside of that.

It can, however, represent changes resulting from long term processes like the shifting of the geomagnetic field, since the IRI uses a modip or geomagnetic coordinate system (depending on the sub-model) and the magnetic field model has been updated over time. In fact, you could try to use the IRI to control against the impacts of geomagnetic field migration in search of climate change impacts, but the model output itself explicitly does not include lower atmospheric climate forcing. The impacts shown in your figures is likely entirely just the impact of the shifting magnetic field and the statistically weak solar activity over the last two cycles. If you ran the model and forced the solar activity term to be constant, you would not see anything other than the geomagnetic field migration impact. Given that you try to remove the MgII forcing later on anyway, there seems to be no reason why you shouldn't just force it to a constant to verify your hypothesis anyway.

Thank you for pointing this out to us.

As you mention, if we keep constant the solar activity index selected in IRI (IRI-Plas, IRI2016 or IRI2020), even though we run the years from 1960 to 2022, there is no solar activity variation in foF2 or hmF2, as can be seen in Figure R8. Instead there is a slow variation (or trend) which corresponds, as you also mentioned, to the Earth's magnetic field inclination angle, which is the one entering the modified dip coordinate used by IRI, and changes with years according to IGRF model. In Figure R8 we have also included foF2 and hmF2 estimated with IRI-2020 default indices (black curves), and use as an input the Rz value of the corresponding dates from 1960 to 2018 (red dashes curves), as you also suggested.

F2 parameters estimated with IRI with the default indices and by entering Rz are very similar, but not identical. We were expecting in hmF2 case to be exactly the same since hmF2 is estimated with Rz in the default option. But it is not the case.

[Figure]

Figure R8. foF2 and hmF2 annual means, at 12 LT, 20°N-30°E, estimated with IRI2020 default parameters (IG in foF2 case and Rz in hmF2 case) (black solid line), by entering Rz for the corresponding dates (red dashed line), and keeping Rz=70 for every date (blue solid line).

Figure R9 shows the trends that we obtain if we run IRI-Plas and also IRI-2020 keeping Rz constant at 70 (that is Rz=70 for every month and year). The trend is assessed directly from the modeled data without any filtering since foF2 does not have any other variability (as seen in Figure R8, blue line), that is

$$foF2 = \alpha\, t + \beta \qquad \text{and} \qquad hmF2 = \alpha\, t + \beta$$

$\alpha$, estimated applying least squares to the regression foF2 vs. t and hmF2 vs t, is the trend. In the case of Figure R9 the trends are estimated for annual mean values of foF2 and hmF2, so t corresponds to years.

[Figure]

Figure R9. Trends estimated with F2 parameters obtained from IRI-Plas (left panels) and IRI2020 (right panels) keeping Rz=70 and running only the years along 1960-2018 (since IRI2020 allows until this year). Black dotted lines indicate trend=0, black dashed line the dip equator in 1960 and black solid line the dip equator in 2022.

Considering Rz fixed at 70, the global mean trends result -0.0004 and -0.0003 MHz/decade for foF2 with IRI-Plas and with IRI2020 respectively. This is almost zero compared with the trends obtained without keeping Rz constant (~-0.10 MHz/decade). In the case of hmF2, with Rz fixed at 70, the mean trends result -0.086 and -0.098 km/decade with IRI-Plas and IRI2020, respectively, that again is almost zero compared with the trends obtained without keeping Rz constant (~-2 km/decade).

This means that, globally, the trend due to the secular variation of the magnetic field inclination cancels out. This is logical since the main change here is due to the displacement of the magnetic equator which induces trends of opposite sign almost symmetrically at each of its sides along its slow displacement.

All this would point out that the trends in our work are obtained due to the filtering "method" (in agreement with our discussion), which means the following: we are filtering with MgII while the interannual time variation of the ionospheric series are determined by another proxy. This implies, as you also correctly noticed to us, that the trend obtained would have nothing to do with an external real forcing. However, in the case of foF2, the proxy determining its interannual variation in IRI is IG, which is obtained from measured foF2 data. Thus, it can be said that foF2 is obtained from measured data assimilated through IG, which is a global index, and then particularized for a location through the CCIR maps. The case of hmF2 is different, as explained in what follows.

In the default mode of IRI2020, foF2 is estimated, as already mentioned, from IG and hmF2 from Rz. This would mean that if we filter the solar activity effect from these ionospheric parameters with IG and Rz respectively, we should expect an ~100% filtering (and thus no trends) except for the effect of the magnetic field (in agreement with your comment) (and equivalent to keeping the solar proxy constant as done with Rz=70).

In the case of IRI-Plas, even though we selected MgII as the solar activity proxy, the procedures that adjust the other proxies in the subroutines ends in foF2 variability being determined by IG and hmF2 by Rz, as in the case of IRI2020, but with slight changes that depend on the proxy selected.

In addition, if we make the difference between trends estimated with MgII and trends estimated with IG in foF2 case (with Rz in hmF2 case), we should be left with the trends that are not due to the magnetic field. In this, we are making the following hypotheses:

1) foF2 ionosonde data interannual variability is composed of:
**solar activity variability**
**+ a trend induced by the magnetic field**
**+ a trend induced by the greenhouse effect**
**+ a random noise** (inherent to any non-ideal time series).

2) IG interannual variability is composed of:
**solar activity variability**
**+ a trend induced by the greenhouse effect** (since, considering that this index results from foF2 measured data from stations far from the magnetic equator, where

the secular variation of the magnetic field is extremely small, we assume that the trend induced by the magnetic field in this case is zero).

3) MgII interannual variability is composed of:
**solar activity variability very close to the solar activity variability of EUV solar spectral range ionizing the F2-layer ionosphere**.

The interannual variability of foF2 estimated with IRI models is forced by:

* the magnetic inclination obtained from IGRF, and

* IG, which carries with it the information of a "global" greenhouse effect and the solar activity variability effect.

So, when we filter foF2 (obtained from IRI) with MgII, or any other proxy except IG, we are left with the variability of the magnetic field and the greenhouse effect.

In the case of hmF2 instead of IG, Rz is used. And the hypothesis here is:

1) hmF2 ionosonde data interannual variability is composed of:
**solar activity variability**
**+ a trend induced by the magnetic field**
**+ a trend induced by the greenhouse effect**
**+ a random noise** (inherent to any non-ideal time series).

2) Rz interannual variability is composed of:
**solar activity variability but not very close to EUV solar spectral range ionizing the F2-layer, which seems a quasidecadal cycle "falling" down along the years, as in the case of IG.** The difference here is that the "falling" of IG would be due to an external forcing, possible the increasing $CO_2$, and the "falling" in Rz we do not know. In fact, Rz would have varied very close to solar EUV until prior to solar cycle 23. After this solar cycle not anymore.

3) MgII interannual variability is composed of:
**solar activity variability very close to the solar activity variability of EUV solar spectral range ionizing the F2-layer ionosphere**.

The interannual variability of hmF2 estimated with IRI models is forced by:

* the magnetic inclination obtained from IGRF, and

* Rz, which does not carry with it the information of a "global" greenhouse effect. However, the solar activity variability it reflects seems to be composed of the quasidecadal well-known oscillation plus a kind of trend towards the last cycles (in particular de minimum epochs).

So, when we filter hmF2 (obtained from IRI) with MgII, or any other proxy except Rz, we are left with the variability of the magnetic field and a downward trend.

If we filter hmF2 from IRI with Rz, we would be left with "nothing".

All these reasoning is checked with Figures R10 and R11 for foF2 (assessed with IRI2020 (default setting) and IRI-Plas (with MgII) respectively) and Figures R12 and R13 for hmF2 (assessed with IRI2020 (default setting) and IRI-Plas (with MgII) respectively).

[Figure]

Figure R10. Trends estimated with foF2 from IRI2020 with default settings (IG for foF2 and Rz for hmF2). Solar activity is filtered with MgII in the upper-left panel, with IG in the lower-left panel, and with Rz in the upper-right panel. The lower-right panel corresponds to the difference between trends with MgII filtering minus trends with IG values, expecting to obtain trends which are not forced by the magnetic field secular variation. Black dotted lines indicate trend=0, black dashed line the dip equator in 1960 and black solid line the dip equator in 2022.

[Figure]

Figure R11. As in Figure R10 but using IRI-Plas with MgII as solar activity proxy.

[Figure]

Figure R12. As in Figure R10 but for hmF2 using IRI2020.

Figure R13. As in Figure R10 but for hmF2 using IRI-Plas with MgII as solar activity proxy.

We included the filtering with Rz, since it gives very similar results to those when using IG, since both have a like a "level falling" of the two last solar cycle minima. In fact, the correlation coefficients between the solar proxies are the following:

| $r^2$ | MgII | IG | Rz |
|-------|------|-------|-------|
| MgII | 1 | 0.950 | 0.954 |
| IG | | 1 | 0.982 |
| Rz | | | 1 |

That is, IG and Rz are more similar.

The mean correlation between foF2 assessed with IRI (default indices) at 12 LT, annual time series, and each of these solar proxies is:

| $r^2$ (global mean) | MgII | IG | Rz |
|---|---|---|---|
| foF2 (IRI) | 0.961 | 0.998 | 0.980 |
| hmF2 (IRI) | 0.969 | 0.983 | 0.972 |

The best correlation in the case of foF2, as expected, is that with IG, since it is this proxy which determines foF2 interannual variability.

However, in the case of hmF2, the highest correlation is also with IG, even though its interannual variability is determined by Rz.

Sorry for being repetitive with all this, but we would like to add the following:

The downward trend obtained in foF2 and hmF2, statistically speaking, is due to the inter-annual time variation of these F2 region parameters that results from two time series (IG and Rz) which present the last two minima weaker than the previous four. This can be seen in Figure R14 were we plot MgII (black line) and IG (red line) annual means. It is clear that if we use MgII to filter a time series behaving like IG, we will obtain a residual with a downward trend, as also shown in Figure R14. Since IG carries the information from ionosondes, then we can expect that the trends obtained could be a reasonable approach to experimental trends.

Rz happens to vary similar to IG. That is, it presents the last two minima lower than the previous minima. This can be seen in Figure R15 were Rz original data base, Rz from SILSO and Mg II have been plot. We included the old Rz series just to notice that the new Rz has a more pronounced decrease during the last minima. We assume that this is the Rz used by IRI. It is clear also here that if we use MgII to filter a time series behaving like Rz, a downward trend will be obtained in the residuals of this regression. And this is why we obtain here a downward trend again. In this case, unlike the foF2 case, it is due to a coincidence: the downward trend of hmF2 from IRI is due to the downward trend of Rz, and that of hmF2 from ionosondes is due to the greenhouse effect (if we assume this is the main forcing). And of course, the downward trend in Rz has nothing to do with the greenhouse effect. They both just happen to be in the same direction.

[Figure]

(a)                                                          (b)

Figure R14. (a) Mg II (black) and IG (red) annual mean time series for the period 1960-2022. (b) Residuals (solid black line) from the fitted regression IG = A MgII + B, together with the residual linear trend (dashed black line).

[Figure]

Figure R15. 12-month running mean series of Rz orginal data (red line), Rz from SILSO (black dashed line) and Mg II (green line).

One more comment. It is curious that Rz seems much better than MgII. It even contradicts the most recent results recommending MgII and F30, followed by F10.7. This is due to our series begin in 1960, while the other papers favoring MgII and/or F30 begin in 1976. However, we think this deserves another deep analysis for another work, or repeating all this considering different sub-periods.

In the revised version of our work we will include now a thorough description of how foF2 and hmF2 are estimated by IRI model, and also the interpretation of the trends will be clearly stated.

Thank you again for your observations and for having the time to meet with us to discuss about our results and their interpretation.

Hoping to meet all your requirements,

Bruno S. Zossi, Trinidad Duran, Franco D. Medina, Blas F. de Haro Barbas, Yamila Melendi, and Ana G. Elias

---

## Referee Report (RR1)

Authors responded adequately to my comments and made adequate changes in the paper. Also their responses to comments of the other reviewer and D. Themens seem to be adequate. Therefore I can recommend publish this paper as it is.

---

## Author Response (AR2)

Dear Editor,

We considered all the technical corrections suggested, which are the following:

line 31: "...for the correct reasons in the hmF2 case."

Done.

line 34: "...and of experimental trends only in the foF2 case."

Done.

line 48: "...the limited time span and sparse spatial ..."

Done.

line 120: "...and Lyman-α, for example, ..."

Done.

line 204: "...being efficiently described by the 12-month..."

Done.

line 206: "...they are all more alike when the time series compared consist of annual means, rather than monthly or daily means. This is because at these shorter timescales, ..."

Done.

line 206: "... averages essentially to zero."

Done.

line 216: "...are 7.93 MHz and 308.6 km, similar to the Table 2 values."

Done.

line 241: "...as well as the spatial variation of the trends."

Done.

line 263: "It is important to remark that the trends reported by Solomon et al. (2018) may have resulted in lower values because they ran the simulation."

Done.

line 264: "This would have neglected part of the trend..."

Done.

line 311: "...where we concluded that both F30 and MgII are equally appropriate, but without being..."

Done.

line 340: "An additional topic deserving further research is the ..."

Done.

line 341: "...but we consider it important ..."

Done.

line 342: "...by Cnossen and Richmond (2012), who analyzed..."

Done.

line 349: "Considering how the foF2 and hmF2..."

Done.

line 353: "...like the greenhouse gas concentration increases..."

Done.

line 355: "Hence, the foF2 trends obtained using IRI-Plas model values can be, to a first approximation, attributed..."

Done.

line 362: "...result from the Rz overall..."

Done.

line 364: "...due to greenhouse gas concentration increases."

Done.

line 365: "This is due to the coincidence that both hmF2 from IRI, and hmF2 from measurements..."

Done.

line 366: "...it is the Rz overall downward trend..."

Done.

line 368: "...greenhouse gases during the last decades."

Done.

line 370: "Smaller Rz values since ~2001 have been ..."

Done.

line 386: "...allowing it to be run on modest..."

Done.

line 397: "...we conclude that the IRI model..."

Done.